# Genetic underpinnings of chills from art and music

Giacomo Bignardi[1,2]*, Danielle Admiraal[1], Else Eising[1‡], Simon E. Fisher[1,3‡]

1 Language and Genetics Department, Max Planck Institute for Psycholinguistics, Nijmegen, The Netherlands, 2 Max Planck School of Cognition, Leipzig, Germany, 3 Donders Institute for Brain, Cognition and Behaviour, Radboud University, Nijmegen, The Netherlands

‡ These authors jointly supervised to this work.
* giacomo.bignardi@mpi.nl

## Abstract

Art can evoke strong emotional responses in humans. Here, we examine genetic contributions to chills, a marker of such responses. We gather self-reports from a genotyped sample of thousands of partly related individuals from the Netherlands ($n = 15,606$). Using genomic relationships based on common single-nucleotide polymorphism (SNP) data, we find that up to 29% of the variation in proneness to aesthetic (visual art and poetry) and music chills can be explained by familial relatedness effects, one-fourth of which is attributed to SNP variation. Furthermore, we reveal a moderate genetic correlation of .58 between aesthetic and music chills, pointing to shared genetic variation affecting susceptibility to strong emotional responses across different art forms. Finally, we find that a polygenic index (PGI) for openness to experience ($n = 220,015$) is associated with susceptibilities to both aesthetic and music chills. Our results show that additive genetic variation, but also familial relatedness beyond shared common SNPs, contributes to proneness to chills from artistic, poetic, and musical expressions. These results open up a promising path towards studying the human attitude towards art, via both state-of-the-art genomics and intergenerational models of transmission.

## Author summary

Many people experience chills when listening to music, reading poetry, or viewing art. Yet not everyone feels these reactions in the same way. These differences provide a window into how our brains and bodies respond to art, revealing individual variation in emotional experiences. To investigate what drives these differences, we analysed data from over 15,500 participants with available genetic information, examining whether DNA variation could help explain why some people are more prone to these intense responses. We estimated that roughly 30% of the variation in chills is linked to family-related factors, of which about

**Data availability statement:** Data may be obtained from a third party and are not publicly available. Researchers can apply to use the Lifelines data used in this study. More information about how to request Lifelines data and the conditions of use can be found on their website https://www.lifelines-biobank.com/researchers/working-with-us.

**Funding:** The Lifelines initiative has been made possible by subsidy from the Dutch Ministry of Health, Welfare and Sport, the Dutch Ministry of Economic Affairs, the University Medical Center Groningen (UMCG), Groningen University and the Provinces in the North of the Netherlands (Drenthe, Friesland, Groningen). G.B. was partly supported by the German Federal Ministry of Education and Research (BMBF) and the Max Planck School of Cognition; G.B., D.A., E.E. and S.E.F. were supported by the Max Planck Society. E.E. is also supported by a Veni grant from the Dutch Research Council (NWO; VI.Veni.202.072).

**Competing interests:** The authors have declared that no competing interests exist.

one-fourth was attributable to common DNA variants. Some genetic influences appear to be shared across music, poetry, and art, and are associated with individual openness to experience, including general artistic interests, while others may be unique to each domain. These results suggest that genetics contributes to how strongly people respond to cultural experiences and pave the way for future studies on the genetics of sensitivity to art and music experiences.

## Introduction

Darwin experienced them while hearing the anthem in King's College Chapel, which gave him "intense pleasure so that [his] backbone would sometimes shiver"([1], p. 61). Nabokov elevated them as "the telltale tingle" needed to bask in the book of genius [2]. When people encounter art, they may experience intense physiological responses that coincide with peak subjective pleasure and emotions, described as "chills" [3]. Since chills are clear, measurable events [4] that link subjective responses with physiological manifestations [4–7], they serve as a model for studying responses to art [3,6–11]. Investigations of chills have revealed that the pleasure evoked by music and poetry recruits neural systems similar to those involved in the evaluation of other biologically relevant non-abstract stimuli [3,7] and have linked stable subjective differences to inter-individual variability at both physiological and neurobiological levels [8,10,12].

Here, we investigate the genetics behind individual differences in proneness to chills, which we suggest can offer an additional approach for empirically studying how humans respond to art. As with many other complex human traits [13], proneness to chills is partly heritable [14]. For example, across diverse cultures, proneness to chills from visual art and poetry (referred to as aesthetic chills) has moderate heritability ($h^2$). Genetic effects have been estimated to account for 36–43% of the variance in large twin-based samples—including more than 10,000 individuals from the Netherlands [14], and thousands from Germany, Denmark, North America, Australia, and Japan [15]—with the remaining variance attributed to residual effects, such as environmental influences and measurement error, with no evidence for environmental effects shared by family members [14]. However, key questions about the genetics of inter-individual differences in proneness to chills remain unanswered. First, to date, $h^2$ estimates have been derived from family studies only, mainly from those relying on the Classical Twin Design (CTD). As such, it is unclear whether previous results reflect molecular genetic effects or biases introduced by unmet assumptions inherent to the CTD. Second, to the best of our knowledge, there is a lack of evidence regarding whether genetic effects on aesthetic chills may extend to other artistic domains, such as music, since the only direct estimates derived so far are specific to chills from visual art and poetry. This gap raises additional questions of whether sensitivity to different art forms is shaped by entirely different processes or is tied to similar underlying biological mechanisms. Finally, it is still unclear whether the genetics of proneness to chills is indeed an index of broader predispositions towards art, or if it reflects an independent phenomenon.

To resolve these uncertainties, we used a large sample of more than fifteen thousand genotyped individuals from Lifelines, a large multi-generational cohort study of individuals living in the North of the Netherlands [16], from whom we gathered self-reports on their proneness to aesthetic and music chills. We applied a molecular approach to quantify $h^2$ that contrasts patterns of measured genotypic resemblance with phenotypic resemblance in unrelated and related individuals [17,18]. The strength of this approach relies on the possibility of partitioning family-based (also called pedigree-based) estimates for $h^2$ ($h^2_{PED}$) into two types of $h^2$. The first, obtained from unrelated and distantly related individuals and referred to as the Single-Nucleotide Polymorphism heritability ($h^2_{SNP}$), estimates the additive effect of genetic variants common in the population under study, captured using DNA genotyping chips. The second, obtained instead from related individuals, assesses additional additive effects of genetic variants untagged by genotyping chips, as well as biases due to, for instance, non-additive genetic and environmental effects shared between members of families [19,20]. The gap between such $h^2_{SNP}$ and $h^2_{PED}$ estimates is sometimes referred to as the (still) missing heritability [21]. Moreover, by collecting information on people's proneness to chills evoked by different art modalities, we could investigate similarities and differences between responses to diverse forms of artistic, poetic and musical expression. We accomplished this by estimating genetic correlations ($r_g$) between aesthetic and music chills. Finally, we investigated the degree to which proneness to chills might reflect a broader genetic predisposition towards art. To do so, we leveraged the largest published genome-wide association study (GWAS) on personality to date [22] to construct an openness to experience polygenic index (PGI), including artistic interests and active imagination, testing their association with proneness to chills in our target sample.

## Results

Working with Lifelines [16], we gathered self-report data on proneness to chills in response to diverse art forms from 35,114 adults (21,249 women). Some 15,615 of these individuals (9,907 women) had genotype data available, and their ages covered the entire adult lifespan, ranging from 18 to 96 years old. Proneness to aesthetic and music chills was assessed using the self-reported items "Sometimes when I am reading poetry or looking at a work of art, I feel a chill or wave of excitement" [23] and "I sometimes feel chills when I hear a melody that I like" [24], respectively. Participants were asked to report how much they agreed with these statements by choosing one out of five response options ranging from "strongly disagree" to "strongly agree". Within-trait descriptive statistics and between-trait phenotypic similarities and differences are reported in the S1 Text.

To assess the extent to which genetic influences account for variation in proneness to aesthetic and music chills, we applied a linear mixed model with a Genome-based Restricted Maximum Likelihood (GREML) estimator [18]. This method estimates the amount of the observed phenotypic variance that can be explained by genomic relatedness between individuals, as indexed by SNP-derived Genetic Relatedness Matrices (GRM). We focused our analysis on a total sample of 15,606 individuals, removing a total of 9 individuals from pairs with estimated genomic relatedness ($\pi$) larger than .90 (i.e., identical twins, see Table A in S1 Text). Taking advantage of the family relatedness structure of the participants included in the Lifelines cohort, we constructed two complementary SNP-derived GRM, one representative of $\pi$ across all 15,606 individuals and another for which $\pi$ below a set threshold (i.e., $\pi < .05$, the recommended genetic relatedness threshold for GREML-based analysis in family data [19], see also S1 Text) was adjusted to zero [17]. Using this approach, also known as threshold GREML [25], we could partition $h^2_{PED}$ into $h^2_{SNP}$ and $h^2_{\pi \geq .05}$, the latter assessing effects beyond common SNP variation tagged by genotyping array (see Fig A in S1 Text). As shown in Table 1, GREML-based analysis of adjusted two-stage rank normalised and residualised data for age, sex, genotyping array, and ten genomic principal components (PC) indicates significant $h^2_{SNP}$ for aesthetic ($\chi^2(1) = 6.30$, $p = .006$) and music ($\chi^2(1) = 9.25$, $p = .001$) chills, with $h^2_{PED}$ explaining up to 29% of the total variance in proneness to chills. These findings indicate that the proneness to chills from art, poetry and music is influenced by genetic variation. Results obtained from raw, transformed data, and by using a different threshold ($\pi < .02$), were similar (see Tables B and C in S1 Text).

**Table 1. Overview of estimates.**

| Component of interest | Parameter | Trait | Estimate | 95% CI |
|---|---|---|---|---|
| Trait variance explained by tagged common DNA variation (with additive effect) | $h^2_{\text{SNP}}$ | Aesthetic chills | .06 | [.01, .10] |
| | | Music chills | .07 | [.03, .11] |
| Gap between effects of common DNA variation and familial relatedness | $h^2_{\pi \geq .05}$ | Aesthetic chills | .18 | [.10, .27] |
| | | Music chills | .23 | [.14, .31] |
| Trait variance explained by overall effects of familial relatedness | $h^2_{\text{PED}}$ | Aesthetic chills | .24 | [.17, .32] |
| | | Music chills | .29 | [.22, .36] |
| Phenotypic correlation | $r$ | Aesthetic-music chills | .43 | [.42, .44] |
| Genetic correlation | $r_g$ | Aesthetic-music chills | .58 | [.20, .95] |
| Percentage of variance explained by PGI for openness to experience | $r^2_{\text{PGI}}$ | Aesthetic chills | 0.3% | [0.1%, 0.4%] |
| | | Music chills | 0.1% | [0.0%, 0.3%] |
| Partner phenotypic correlation | $r_{\text{AM}}$ | Aesthetic chills | .13 | [.10, .16] |
| | | Music chills | .11 | [.08, .15] |
| Cross-trait partner phenotypic correlation | $r_{\text{xAM}}$ | Aesthetic-music chills | .03 | [-.01, .06] |

*Note*: $h^2_{\text{SNP}}$: GREML-SNP-based heritability; $h^2_{\pi \geq .05}$: GREML-based excess heritability in related individuals; $h^2_{\text{PED}}$: GREML pedigree-based heritability; $r$: Pearson-correlation; $r_g$: GREML-SNP-based additive genetic correlation; PGI: Polygenic Index; $r_{\text{AM}}$: within-trait Pearson-correlation between partners; $r_{\text{xAM}}$: cross-trait Pearson-correlation between partners. All results shown here were obtained from fully adjusted two-stage rank normalisation procedure (rank transformed) on the phenotype data, with exception of phenotypic correlations ($r$), which used non-adjusted scores. 95% Confidence Intervals (CI) for heritability and genetic correlation estimates are derived from the Standard Errors (i.e., 95% CI = estimate ± 1.96*SE).

Since both aesthetic and music chills displayed significant $h^2_{\text{SNP}}$, we assessed evidence for shared genetic contributions to variability in the two traits. First, we estimated the phenotypic correlation between proneness to aesthetic and music chills. To obtain a more precise estimate, we utilised the full Lifelines sample of 35,114 individuals with available proneness to chills data. Proneness to aesthetic and music chills (raw scores) correlated modestly, with a $r$ of .43 (95% Confidence Interval (CI) [.42, .44]). Then, we applied a bivariate extension of GREML to estimate the genetic correlation between the proneness to aesthetic and music chills in the remaining genotyped sample of 15,606 individuals. We found a moderate genetic correlation, with an $r_g$ of .58 (95% CI [.20, .95], $\chi^2(1) = 4.61$, $p = .02$; which was also significantly lower than 1, $\chi^2(1) = 4.17$, $p = .02$, both one-tailed tests). These results suggest that a substantial proportion of the genetic variation associated with proneness to aesthetic chills is also associated with music chills. Yet, they also reveal some degree of specificity in genetic contributions to the different traits.

Finally, we assessed whether the genetics of proneness to chills could in part be reflected by broader genetic associations with art interest. We derived individual PGI from the largest published genome-wide association study of openness to experience, which was previously conducted in a fully independent sample of 220,015 individuals [22]. To construct the PGI, we used SNP associations with openness to experience, assessing artistic interests and active imagination (see Methods). We found that PGI were significantly associated with proneness to aesthetic ($\beta = .05$, $p = 4.79 \times 10^{-10}$) and music chills ($\beta = .03$, $p = 1.97 \times 10^{-6}$), explaining 0.3% ($r^2_{\text{PGI}}$ 95% CI [0.01%, 0.4%]) and 0.1% ($r^2_{\text{PGI}}$ 95% CI [0.0%, 0.3%]) of the overall variance, respectively. Similar results were obtained in a subset of 10,703 distantly related and unrelated individuals ($\pi < .05$; See S1 Text). Further analyses provided no evidence for heterogeneous PGI associations with either aesthetic or music chills ($Q_{\text{PGI}} = 1.95$, $p = .10$), suggesting that PGI explained about 0.5% (95% CI [0.2%, 0.8%]) of the overall covariance between the two traits (see Methods, S1 Text, and Fig B in S1 Text). Thus, genetic variation associated with individual differences in openness to experience, including art interest, contributes to proneness to chills across art domains. Table 1 provides a comprehensive overview of the results of these different investigations.

We note that assortative and cross-assortative mating can lead to bias when estimating $h^2_{\text{SNP}}$ and $r_g$. Assortative and cross-assortative mating refer to phenomena where two individuals are more likely to mate when they resemble each

other in one or more (e.g., cross) traits. These phenomena have been shown to confound $h^2_{SNP}$ and $r_g$ by upwardly biasing estimates [26,27]. To provide evidence of minimal upward bias on $h^2_{SNP}$, and help solidify the interpretation of $r_g$ as an estimate for widespread pleiotropy unconfounded by both types of assortment, we estimated trait and cross-trait partner correlations in proneness to chills. To do so, we studied 3408 partners for whom phenotypic data were available. We observed small yet significant partner correlations for aesthetic chills, $r_{AM}$ = .13 (95% CI [.10, .16]), and music chills, $r_{AM}$ = .11 (95% CI [.08, .15]). There was no evidence of cross-trait partner correlation, with a non-significant correlation of only .03 (95% CI [-.01, .06]). These findings indicate that assortative and cross-assortative mating do not largely bias our estimates for $h^2_{SNP}$ and $r_g$ (for details, see S1 Text and reference [26]).

## Discussion

Our results provide converging evidence for the existence of a genetic predisposition to the ability to experience chills from cultural products, such as artistic, poetic, and musical expression. By investigating the similarity between genotypic and phenotypic resemblance in related and unrelated individuals, we find significant SNP-based heritability for aesthetic and music chills, aligning with the claim that proneness to chills may represent a "vector of biological variation" [28]. Consistent with other studies [14,15], the substantial proportion of variance attributable to residual effects—including environmental influences, measurement error, but also stochastic developmental processes unique to each individual—indicates that factors beyond genetic variation also contribute to individual differences in proneness to chills. Furthermore, we reveal a substantial genetic correlation between the traits, which indicates that proneness to chills from different art modalities is partly tied to shared genetic variation. In line with these findings, we also find that PGI for openness to experience, including artistic interests, are significantly associated with proneness to aesthetic and music chills. At the same time, the genetic correlation between propensity to aesthetic and music chills is significantly lower than 1, indicating less than full overlap in the underlying genetic architecture for responses to different forms of artistic expression.

Notwithstanding the significant SNP-based heritability, this explained only around a fourth of the total pedigree heritability and a fifth of the previously reported twin-based heritability for aesthetic chills (obtained from another sample of the Dutch population [14]). While in line with well-established concepts of "missing heritability" [29], as SNP-based estimates tend to explain only one-third to one-half of family-based estimates of complex human traits [20,21,30], the gap observed here for proneness to chills seems relatively large. (Note, however, that this is still smaller than the gap observed for some other behavioural characteristics, for which $h^2_{SNP}$ is only about one-tenth of the $h^2_{TWIN}$; see reference [30] for an example). The observed gap between SNP- and family-based heritability estimates likely involves a complex mixture of factors. These include contributions of imperfect tagging of causal variants from genotyping arrays, such as rare variants, non-additive genetic effects, such as dominance deviations or higher-order epistasis (gene-by-gene interaction at different genetic loci), and common environmental effects shared across members of families [17]. Additional contributors may include unequal environmental sharing among relatives and interactions between genes and shared environmental factors [31].

It is worth noting here that while $h^2_{SNP}$ estimates can be considered the upper bound of the additive genetic influences that genome-wide association studies (GWAS) could capture on the same genotyping platform, they are generally expected to underestimate the true heritability of the trait in the population under study. This discrepancy arises because the GRM imperfectly captures the contribution of ungenotyped variants among unrelated or distantly related individuals. By contrast, when the same GRM is obtained from pairs of individuals above a certain threshold of relatedness (e.g., $\pi \geq$ .05), the estimates tend to be closer to the true heritability. This is because, at higher levels of relatedness (larger values of $\pi$), the GRM better approximate the genetic covariance across the genome. Therefore, a gap between $h^2_{SNP}$ and $h^2_{PED}$ is expected [17]. At the same time, the $h^2_{PED}$ estimates derived from related individuals can be upwardly biased if shared environmental effects, non-additive genetic effects, or effects of rare variants are present [32]. Previous CTD studies on proneness to aesthetic chills found no evidence of shared environmental effects [14,15], with correlations among

monozygotic twins exceeding twice those of dizygotic twins [14]. These findings, under the assumptions of identical shared environmental influences in monozygotic and dizygotic twins and no gene-by-shared environment interaction, suggest that non-additive genetic effects, rather than shared environmental effects, may partly contribute to the gap between $h^2_{SNP}$ and $h^2_{PED}$ as found in this study. However, an alternative explanation, which cannot be ruled out in the current study, is also possible: gene-by-shared environment interactions. If gene-by-shared environment interactions were to influence phenotypic variation, then both the current $h^2_{PED}$ and the previously reported $h^2_{TWIN}$ may have yielded upwardly biased estimates of $h^2$ [33]. Furthermore, although we noted that the observed weak correlation between partners was not sufficient to largely bias $h^2_{SNP}$ estimates in this study, it is still possible that assortment may have previously upwardly biased $h^2_{TWIN}$ estimates (as these were obtained from a constrained ADE model, which is expected to provide inflated $h^2_{TWIN}$ estimates under assortative mating [33]). Therefore, assortative mating may still play a role in the gap observed between the estimates provided in this study and the ones reported from previous CTD studies. Future studies with different research paradigms will help to better understand this gap. In particular, more sophisticated twin-based approaches, such as extended family twin designs (in which parents, partners, and siblings of twins are included [33,34]), may be particularly suited to partly explain the gap observed in this study.

In addition, our study reveals the association between proneness to chills and PGI for openness to experience—assessed as individuals' interest towards the arts and self-reported active imagination (see below and Methods). We note that the percentage of variance in proneness to aesthetic and music chills explained by the PGI is less than 1%. However, the modest proportion of variance accounted for here is in line with theory-driven expectations on PGI-based predictions. In particular, it is known that the expected percentage of phenotypic variance that can be explained by a PGI is a function of the heritability of the target phenotype being predicted, the one from which the PGI is derived, and their genetic correlation, as well as the sample size of the original GWAS, along with the effective number of SNPs included [35,36]. Considering the observed $h^2_{SNP}$ of proneness to chills and the previously reported heritability of openness to experience [22] ($h^2_{SNP}$ = .05), then the maximum expected percentage of variance that we could have explained is ~1% (see S1 Text) for aesthetic and music chills, even assuming an unlikely perfect genetic correlation of 1. These expectations are not far from our observed $r^2_{PGI}$, thus contextualising our estimates for the absolute percentage of variance explained by the PGI as being relatively close to theoretical expectations.

This study comes with a few limitations. First, our PGI analyses made use of summary statistics from GWAS efforts conducted with self-reports that included art interest and active imagination [22]. As such, individual PGI assess also other indicators of openness to experience beyond art interest alone. Second, the results in the present study were obtained from a population that was representative of the Netherlands and focused on individuals of European descent. Further studies should assess the generalisability of our findings beyond ancestries of European descent and culture. Indeed, the cross-cultural prevalence of experiencing chills [37] and proneness towards these [23], and the observation that such proneness may rely less on cultural bounds and influences from Western humanistic traditions than other traits [23,28], leads to intriguing questions for future research. In particular, it will be interesting to ask whether proneness to chills may be used to investigate the genetics of human attitudes towards art across cultures. Moderating effects introduced by cultural or generational differences in genetic influences could be an especially promising avenue of study [38]. Third, our results do not provide insights into how molecular associations with proneness to chills may extend to biological mechanisms underlying related physiological processes, such as piloerection (i.e., hairs standing on end). Previous research suggests that piloerection tends to co-occur with the experience of chills [39], although this is not always the case [40]. At the same time, the findings of genetic associations with proneness to chills may allow future research to better disambiguate between these two seemingly tied phenomena. Fourth, sampling bias in the Lifelines cohort may affect heritability and genetic correlation estimates. While Lifelines represents the northern Netherlands well, women, middle-aged individuals, and individuals with higher educational attainment tend to be slightly overrepresented [41]. Since proneness to chills is linked to both demographics and education [42], this may introduce bias in genetic estimates. However, evidence from

other biobanks suggests that heritability is typically only modestly underestimated under such bias [43], with downstream analyses being more strongly affected [44]. Therefore, genome-wide association studies may be more susceptible to this type of bias than the present study. Future work should address this issue, for example, by applying weighting to genetic associations based on census data on sex and educational attainment, to better align the sample estimates to the effects in the overall population [45]. Fifth, the items used to assess proneness to chills are likely subject to measurement error. For instance, the test-retest over a 5-year timespan for aesthetic chills in adults has been previously estimated to range between .58 and .61, in a comparable but not overlapping Dutch sample [14]. As such, our $h^2$ estimates are likely downwardly biased by intra-individual variance (including measurement error), and therefore should be considered conservative lower bounds for the additive effects of SNP variation.

In conclusion, our research highlights shared molecular heritability for proneness to chills from art, poetry, and music. Furthermore, it reveals that proneness to chills is associated with a PGI for openness to experience, partly indexing genetic associations with art interest. Yet it also reveals larger estimates for genetic effects on proneness to chills in related individuals, hinting at deviations from pure additive genetic transmission of sensitivity to art and underscoring the need for extended intergenerational designs to relax classic modelling assumptions. We anticipate that larger-scale GWAS of proneness to chills, combined with extended family designs, could further illuminate the origins and biological bases of affective responses to art. Such studies would enable the exploration of both molecular and cultural processes and may even clarify whether the aetiology of these subjective experiences differs from that of other sensory experiences [46].

## Materials and methods

### Ethics statement

The general Lifelines protocol has been approved by the University Medical Center Groningen medical ethical committee (2007/152), and the "Speech, Language and Musicality" study was reviewed by the medical ethical committee of the Erasmus Medical Center (MEC-2022–0313). All participants gave written informed consent prior to taking part in the survey. More information on how informed consent was obtained can be found at https://wiki.lifelines.nl/doku.php?id=informed_consent.

### Sample

Individuals included here participated in the Lifelines study "Speech, Language and Musicality". Lifelines is a multi-disciplinary prospective population-based cohort study examining in a unique three-generation design the health and health-related behaviours of 167,729 persons living in the North of the Netherlands. It employs a broad range of investigative procedures in assessing the biomedical, socio-demographic, behavioural, physical, and psychological factors which contribute to the health and disease of the general population, with a special focus on multi-morbidity and complex genetics [16]. The study population included in Lifelines is broadly representative of the adult population of the north of the Netherlands. Participants in Lifelines were invited by their General Practitioner (GP) to be part of the original study. Family members of individual participants, including partners, were invited by the participants themselves to take part in the study. Family relations data in Lifelines come from municipal registries. The UMCG Genetics Lifelines Initiative (UGLI) consortium refined these using anonymised surname data and family composition questionnaires, then optimised them with genetic data. Couples living together without children were also registered as partners within the same family. Documentation of the reconstruction and quality control process is available at https://wiki-lifelines.web.rug.nl/doku.php?id=family_relations. Partner information was based on baseline data (i.e., between 2007 and 2013). Lifelines had no specific selection criteria for exclusion, but GPs were asked to decide which individuals to invite based on a list of possible criteria provided by Lifelines. The list can be found at https://wiki-lifelines.web.rug.nl/doku.php?id=cohort.

Between September and October 2022, approximately 115,000 adult Lifelines participants were asked to participate online in the "Speech, Language and Musicality" study, of whom 35,179 completed the questionnaire. The response rate observed here (~ 30%) is consistent with prior musicality research in national registries. For instance, studies of music, art, and cultural engagement in the Swedish Twin Registry reported comparable rates of participation of 29–36% [47]. It is also worth noting that previous work found no evidence of sampling bias for music reward sensitivity, a construct that encompasses proneness to chills [47].

## Phenotyping

Aesthetic and music chills were measured using validated items in the revised NEO personality inventory (NEO PI-R(23)) and the Barcelona Music Reward Questionnaire (BMRQ [8]). The questionnaires, which were administered online, were collected in Dutch. The sample of participants for which we had data available on both items was 35,114 individuals. This included a subsample of 15,615 individuals for whom we had genome-wide SNP genotype data available. From this subset, we additionally removed 9 individuals from pairs with $\pi > .90$, resulting in a final sample of 15,606 individuals. The sample stratified by $\pi$ can be found in Table A in S1 Text.

## Genotyping

Genotyping was carried out by using the Infinium Global Screening Array for 9251 individuals (release 1, version 2) and the FinnGen Thermo Fisher Axiom for 6364 individuals (release 2, version 2). Information about quality control of geno-typed data and samples, genetic imputation, and genomic PC calculations can be found at https://wiki.lifelines.nl/doku. php?id=ugli. For the individual genotypes obtained using the FinnGen Thermo Fisher Axiom array (release 2, version 2), filtered non-imputed genotyping data were pruned and then used to calculate principal components using PLINK (version 1.9). Further information can be found in the S1 Text file.

## Covariates

Following de Hoyos et al. [48], data were transformed before model fitting using a fully adjusted two-stage rank normalisation procedure [49]. Age, sex, genotyping array, and the first ten genomic PC were the covariates of interest. Results from untransformed raw data are included in Table B in S1 Text.

## Phenotypic analysis

Given that the full sample comprised both related and unrelated individuals, the standard errors (SE) and, by extension, the 95% confidence intervals (CI) surrounding the estimated correlation coefficient are expected to be downwardly biased [50]. To address this limitation and obtain robust estimates of uncertainty for the point estimate, individuals were clustered within families using the cluster argument in the sem() function of the lavaan [51] R package (i.e., sem(..., cluster = "FAM_ID")).

More information on how Lifelines assigns individuals' family IDs can be found at https://wiki-lifelines.web.rug.nl/doku. php?id=family_relations.

## Genome-based restricted maximum likelihood

Estimates for $h_{\mathrm{SNP}}^2$, $h_{\pi \geq .05}^2$, and $h_{\mathrm{PED}}^2$ of aesthetic and music chills were obtained by using a threshold GREML approach [17,18] as implemented in the GCTA software (https://cnsgenomics.com/software/gcta/), version 1.93.2beta [18]. Previous GREML studies use $\pi$ thresholds of .025 and .05, which define the separation between distantly and closely related individuals. We followed the standard of .05 (recommended value [19]) and additionally used a stricter $\pi < .02$ threshold to enhance robustness, based on evidence of possible inflation between $\pi$ and trait resemblance for $.02 < \pi < .05$ [20].

Estimates obtained using a more stringent threshold of $\pi < .02$ are provided in Table C in S1 Text. Similarly, a bivariate extension of GREML [52] was used to obtain estimates of the covariances between the variance components of the two phenotypes, which were then used to obtain estimates for $r_g$. To test whether $h^2_{\text{SNP}}$ and $r_g$ estimates significantly differ from 0 and 1, GCTA applies a Likelihood Ratio Test comparing the model in which $h^2_{\text{SNP}}$ and $r_g$ are freely estimated to a model in which $h^2_{\text{SNP}}$ and $r_g$ are fixed to 0 and 1, respectively. To control for two comparisons, we used a more stringent $\alpha$ of .025. Schematic representations of the GREML approaches are given in Fig A in S1 Text. The extended method section can be found in the S1 Text file. Robustness analyses with meta-analytic estimates obtained from subsamples corresponding to the two UGLI genotyping releases are provided in Table D in S1 Text. We note that, $h^2_{\pi \geq .05}$ being equal to $h^2_{\text{PED}} - h^2_{\text{SNP}}$, $h^2_{\pi \geq .05}$ and $h^2_{\text{PED}}$ are sometimes used interchangeably in the literature to describe pedigree-based heritability. The latter nomenclature is appropriate when $h^2_{\pi \geq .05}$ and $h^2_{\text{SNP}}$ are not estimated simultaneously, as in reference [20].

### Polygenic Index analyses

We computed individuals' Polygenic Index (PGI) for openness to experience using summary statistics from a GWAS that totalled 220,015 individuals [22]. Openness to experience sum scores were derived from only two self-reports: "I see myself as someone who has few artistic interests" (reverse coded) and "I see myself as someone who has an active imagination" (five response options, "disagree strongly" to "agree strongly"). The summary statistics were filtered, removing ambiguous, duplicated and rare (MAF < 0.01) SNPs. PGI were calculated using PRS-CS (version Jan 4 2021) [53] in PLINK, using a European linkage disequilibrium panel from the 1000 Genomes. Genotyping data of individuals with proneness for chills data remained for PGI analyses. We then regressed proneness to aesthetic or music chills on standardised PGI. The $r^2_{\text{PGI}}$ estimates were derived from a PGI regression with robust standard error, as in reference [54]. Supplementary analyses with retained unrelated or distantly related individuals, corresponding to $\pi < .05$ are included in the S1 Text. The bivariate PGI heterogeneity statistic, $Q_{\text{PGI}}$ (i.e., $\chi^2(1)$), as well as the amount of covariance explained by the PGI, was obtained via an adapted bivariate PGI Common Pathway Model [55,56]. More details are provided in the S1 Text file.

## Supporting information

**S1 Text. Supporting information for Genetic underpinnings of chills from art and music.**
(PDF)

## Acknowledgments

The authors wish to acknowledge the services of the Lifelines Cohort Study, the contributing research centres delivering data to Lifelines, and all the study participants. We further wish to thank MacKenzie D. Trupp for feedback on an earlier version of the manuscript, Lucia de Hoyos for sharing the R script to apply the fully adjusted two-stage rank normalisation procedure, and Henkjan Honing for assisting with the Dutch translation of the music chills item.

## Author contributions

**Conceptualization:** Giacomo Bignardi.

**Formal analysis:** Giacomo Bignardi, Danielle Admiraal.

**Funding acquisition:** Else Eising, Simon E. Fisher.

**Investigation:** Giacomo Bignardi.

**Methodology:** Giacomo Bignardi.

**Project administration:** Else Eising.

**Supervision:** Else Eising, Simon E. Fisher.

**Visualization:** Giacomo Bignardi.

**Writing – original draft:** Giacomo Bignardi.

**Writing – review & editing:** Giacomo Bignardi, Danielle Admiraal, Else Eising, Simon E. Fisher.

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
