## [Decision Letter · Decision Letter 0]

20 Jun 2025

PGENETICS-D-25-00453

Genetic underpinnings of chills from art and music

PLOS Genetics

Dear Dr. Bignardi,

Thank you for submitting your manuscript to PLOS Genetics. After careful consideration, we feel that it has merit but does not fully meet PLOS Genetics's publication criteria as it currently stands. Therefore, we invite you to submit a revised version of the manuscript that addresses the points raised during the review process.

Please submit your revised manuscript within 60 days Aug 19 2025 11:59PM. If you will need more time than this to complete your revisions, please reply to this message or contact the journal office at plosgenetics@plos.org. Please include the following items when submitting your revised manuscript:

We look forward to receiving your revised manuscript.

Kind regards,

Zihuai He

Academic Editor

PLOS Genetics

Santhosh Girirajan

Section Editor

PLOS Genetics

Aimée Dudley

Editor-in-Chief

PLOS Genetics

Anne Goriely

Editor-in-Chief

PLOS Genetics

**Journal Requirements:**

https://journals.plos.org/plosgenetics/s/submission-guidelines#loc-parts-of-a-submission

- ® on page: 7.

5) We have noticed that you have uploaded Supporting Information files, but you have not included a list of legends. Please add a full list of legends for your Supporting Information files after the references list.

**Reviewers' comments:**

Reviewer's Responses to Questions

**Comments to the Authors:**

Reviewer #1: Thank you for the opportunity to review this interesting manuscript. This paper reports a novel genomic analysis of proneness to chills, including aesthetic and music chills and their overlap. The paper was very well-written and the methodology was rigorous. I have a few comments, but I don’t view any of them as major concerns, so I support the publication of this manuscript.

Abstract: “data for more than 100 million pairs of individuals” is kind of misleading since you already stated the sample size and there are only 15k unique individuals in the dataset. I think you should remove this phrase.

Method: Although you may be restricted by the word limit, I was hoping to see a little more basic information about the Lifelines study, including when the sample was recruited, how the questionnaires were administered, and whether there is any reason to expect selection bias played a role in these results (e.g., based on which subsets of subjects completed the BMRQ and where genotyped, both of which were much smaller than the total sample).

More importantly, it would be helpful for the authors to be clearer about how many related individuals are included in the final sample. I’m flexible on how to approach this, but it would nice to at least know how many 1st or 2nd degree relatives comprised the final 15k sample. Are there any twins in this dataset? Some information is reported in the PGI section (see comment below) but I think it should be mentioned in a different place than the PGI section.

PGI analyses: why exclude individuals related greater than the 3rd degree? These individuals all have unique PGIs and phenotype data, and you could cluster the data by family using standard approaches that would adjust standard errors for such clustering. My understanding is that this would enable the use of all possible data in these analyses and should therefore result in more accurate estimates (especially since it’s about 4k extra individuals!). However, if there is a reason to avoid including related individuals in this analysis, please explain.

“The remaining GWAS cohorts that contributed to the meta-analyses included summary statistics from OE scores derived from 12 to 48 items…” I don’t see the acronym “OE” described elsewhere so I think this needs to be explained. It was also unclear if data from these individuals were included in the PGI or not. Please clarify.

Finally, I worry that the description of the PGI as ‘genetic predispositions towards art’ is not entirely accurate. I appreciate the authors efforts to hone in on items from the openness to experience GWAS and the reverse coded item about artistic interests makes a lot of sense. I’m just not sure I agree that describing yourself as “someone who has an active imagination” is concretely linked to predisposition towards art (e.g., many artists work only from models or what they are seeing, etc.). I suggest either removing the second item or adding some further justification that these items are really genetically linked (e.g., a strong genetic correlation with item #1 would be sufficient). This comment is still considered relatively minor as it is addressed in the limitations paragraph.

Discussion: The other limitations of the study were discussed well, and I found the discussion of the differences between SNP and pedigree heritability very useful. One note is that the heritability estimates are quite high considering that these were based on individual items. You may want to state that these might actually be underestimates of the true heritability (e.g., given that there is likely more measurement error on a single item than if you had looked at an entire questionnaire).

Reviewer #2: The authors of this manuscript aim to explore the genetic underpinnings of "chills," a physiological response often associated with emotional or environmental stimuli. By leveraging heritability analyses and linear mixed models, the authors seek to quantify the genetic contributions to chills and provide insights into their biological mechanisms.

Major Comments

1. Environmental and Cultural Influences on Chills

While the study provides valuable insights into the heritability of chills, the authors appear to underestimate the influence of environmental and cultural factors. Chills are not purely genetically driven; family culture, upbringing, and environmental factors likely serve as significant upstream regulators. The hypothesis attributing chills primarily to genetic factors needs more robust justification.

o It is essential to adjust for environmental factors, such as family culture, in the heritability models.

o The manuscript would benefit from the addition of a “control arm,” comprising individuals from families without such cultural predispositions to chills. This would allow a more direct assessment of genetic versus environmental contributions.

o The authors should explicitly justify their hypothesis by first demonstrating that genetic factors, rather than environmental ones, primarily drive the observed phenomena. Addressing this issue at the outset will strengthen the foundation of the study.

2. Unveiling the Underlying Mechanisms

While the heritability estimates provide a starting point, the manuscript lacks an in-depth exploration of the molecular mechanisms underlying chills. Identifying associated genes, pathways, and biomarkers is crucial for understanding their genetic basis.

o A genome-wide association study (GWAS) or similar approach to identify specific SNPs and pathways associated with chills is strongly recommended.

o Mapping these genetic factors to relevant KEGG pathways or other molecular networks could illuminate potential biological mechanisms and enhance the translational value of the findings.

3. Longitudinal Nature of Chills

Unlike static genetic traits, chills are likely dynamic and influenced by time-varying factors such as age, mood, and environmental conditions. The authors should account for these temporal variations in their analysis.

o A longitudinal study design that tracks changes in the frequency or intensity of chills over time would provide a more comprehensive understanding.

o Confounding variables such as cohort, gender, and environmental exposures should be carefully adjusted in any longitudinal analysis.

o This approach would also help distinguish between genetic predispositions and environmental triggers, further bolstering the study’s findings.

Summary and Suggestions

Overall, this manuscript addresses an intriguing question about the genetic basis of chills. However, it would be significantly improved by incorporating adjustments for environmental factors, expanding the analysis to explore molecular mechanisms, and adopting a longitudinal framework to capture the dynamic nature of chills. These refinements would make the study more robust and enhance its scientific and practical impact.

Reviewer #3: Review is uploaded as an attachment

**Have all data underlying the figures and results presented in the manuscript been provided?**

Reviewer #1: Yes

Reviewer #2: None

Reviewer #3: None

PLOS authors have the option to publish the peer review history of their article (what does this mean? ). If published, this will include your full peer review and any attached files.

**Do you want your identity to be public for this peer review?** For information about this choice, including consent withdrawal, please see our Privacy Policy .

Reviewer #1: **Yes:** Daniel Gustavson

Reviewer #2: **Yes:** Xinran Qi

Reviewer #3: No

**Figure resubmission:**
---

## [Decision Letter · Decision Letter 1]

19 Nov 2025

PGENETICS-D-25-00453R1

Genetic underpinnings of chills from art and music

PLOS Genetics

Dear Dr. Bignardi,

Thank you for submitting your manuscript to PLOS Genetics. After careful consideration, we feel that it has merit but does not fully meet PLOS Genetics's publication criteria as it currently stands. Therefore, we invite you to submit a revised version of the manuscript that addresses the points raised during the review process.

Please submit your revised manuscript within by Dec 19 2025 11:59PM. If you will need more time than this to complete your revisions, please reply to this message or contact the journal office at plosgenetics@plos.org. Please include the following items when submitting your revised manuscript:

We look forward to receiving your revised manuscript.

Kind regards,

Zihuai He

Academic Editor

PLOS Genetics

Santhosh Girirajan

Section Editor

PLOS Genetics

Aimée Dudley

Editor-in-Chief

PLOS Genetics

Anne Goriely

Editor-in-Chief

PLOS Genetics

**Journal Requirements:**

We have noticed that you have uploaded Supporting Information files, but you have not included a list of legends. Please add a full list of legends for your Supporting Information files after the references list.

**Reviewers' comments:**

Reviewer's Responses to Questions

**Comments to the Authors:**

Reviewer #2: I appreciate the authors’ extensive and thoughtful responses to the first-round reviews. The revisions significantly strengthen the manuscript, and many of my earlier concerns have been satisfactorily addressed. The clarification of the Lifelines sampling structure, the expanded methodological explanations for GREML and PGI analyses, and the enriched discussion about environmental influences, assortative mating, and measurement error all improve the clarity and interpretability of the results.

Below I provide a brief assessment of how well each of my original major comments has been addressed, followed by a few remaining issues that would still benefit from clarification before publication.

Major Points Addressed

1. Environmental and cultural influences on chills

The authors now clearly articulate that chills are influenced by both genetic and non-genetic factors and provide supporting evidence from prior twin studies demonstrating minimal shared environmental effects. They also explain why conditioning on environmental variables or restricting families would bias heritability estimation. This sufficiently resolves my original concern.

2. Mechanistic (molecular) interpretation

While a GWAS was not added, the authors provide a reasonable justification for why the current sample size and design are insufficient for pathway-level work. They appropriately frame this as future work. This is acceptable.

3. Temporal / longitudinal considerations

The new discussion of test–retest reliability, age/sex effects, and how within-person variability biases heritability downward is helpful and sufficient for a cross-sectional genetic design.

Overall, the authors have addressed the primary conceptual concerns.

Remaining Issues That Need Further Attention Before Acceptance

Although the manuscript is now much improved, a few points remain incompletely resolved or still require clarification in the main text (not only in the response letter). Addressing these will ensure proper alignment between the revised manuscript and the authors' explanations.

1. The justification for the PGI interpretation is still somewhat overstated

Although the authors softened the language, the PGI is still described as capturing “openness to experience, including propensities towards art.” The PGI includes multiple openness subdomains, many unrelated to artistic inclination (e.g., imagination, curiosity, unconventionality).

Remaining concern:

The manuscript still risks implying that the PGI specifically indexes “artistic predisposition,” even though this is only one component of the underlying GWAS.

Suggestion:

Clarify in the Results and Discussion that: the PGI captures broad openness-related variation, of which artistic interest is only one facet, and the PGI should not be interpreted as uniquely or predominantly indexing artistic predisposition.

This is already hinted at in the response letter but should be stated explicitly in the manuscript text.

2. The manuscript now uses \pi < .05 as the main relatedness threshold, but the justification still feels fragmented

The response letter provides a thorough explanation, but the main text of the manuscript still does not fully explain:

(a) Why \pi < .05 is now the primary choice

(b) Why \pi < .02 is retained only as a sensitivity analysis

(c) How this aligns with best practices and the cited recommendations

(d) Why GREML estimates are stable across thresholds

Suggestion:

Add one or two clarifying sentences in the Methods summarizing this rationale (instead of relying solely on Supplementary Table S2), so future readers do not need to infer it from the supplementary materials.

3. Slight overstatement of novelty regarding cross-modal genetic correlation

The authors claim there was previously a “complete lack of evidence” for whether genetic effects on aesthetic chills extend to music. While true for molecular genetic studies, prior twin studies (including those by the first author) have already demonstrated overlapping heritable components of emotional response to aesthetic stimuli.

Suggestion:

Rephrase slightly to reflect that the novelty lies in molecular genetic estimation, not the broader behavioral-genetic domain.

4. The discussion of gene-by-shared-environment interaction is new but still somewhat abstract

The manuscript now raises the possibility that G×shared environment interaction could inflate pedigree-based estimates, but the logic remains difficult to follow because:

(a) No empirical test is performed

(b) No specific candidate shared environments are discussed

(c) The implications for interpreting the numeric difference between h²_SNP and h²_ped remain vague

Suggestion:

Add one sentence clarifying the practical takeaway for readers, e.g.: “…although we cannot fully distinguish G×shared environment interaction from non-additive genetic variance using the current design, the stability of estimates across sensitivity analyses suggests that such effects, if present, are unlikely to be large.”

5. The revised manuscript is long and complex; some parts would benefit from clearer figure/table guidance

This is not an error, but a readability refinement. For example:

Supplementary Table S3 (relatedness distribution) is central and could be referenced more prominently.

The PGI analyses (two approaches) could benefit from a short schematic or summary table.

This would greatly help readers unfamiliar with GREML + PGI hybrid designs.

Conclusion

Overall, the authors have substantially improved the manuscript and addressed nearly all major concerns from the first review round. The study is carefully executed, methodologically sound, and provides a valuable contribution to the literature on the genetic architecture of emotional responses to art and music.

Reviewer #3: I have no more questions. All my comments are fully addressed in revised manuscript.

**Have all data underlying the figures and results presented in the manuscript been provided?**

Reviewer #2: Yes

Reviewer #3: Yes

PLOS authors have the option to publish the peer review history of their article (what does this mean? ). If published, this will include your full peer review and any attached files.

**Do you want your identity to be public for this peer review?** For information about this choice, including consent withdrawal, please see our Privacy Policy .

Reviewer #2: **Yes:** Xinran Qi

Reviewer #3: No

**Figure resubmission:**
---

## [Editor Report · Decision Letter 2]

18 Dec 2025

Dear Dr Bignardi,

We are pleased to inform you that your manuscript entitled "Genetic underpinnings of chills from art and music" has been editorially accepted for publication in PLOS Genetics. Congratulations!

Yours sincerely,

Zihuai He

Academic Editor

PLOS Genetics

Santhosh Girirajan

Section Editor

PLOS Genetics

Aimée Dudley

Editor-in-Chief

PLOS Genetics

Anne Goriely

Editor-in-Chief

PLOS Genetics

BlueSky: @plos.bsky.social

Comments from the reviewers (if applicable):

**Data Deposition**

http://datadryad.org/submit?journalID=pgenetics&manu=PGENETICS-D-25-00453R2

**Press Queries**

---

## [Editor Report · Acceptance letter]

PGENETICS-D-25-00453R2

Genetic underpinnings of chills from art and music

Dear Dr Bignardi,

We are pleased to inform you that your manuscript entitled "Genetic underpinnings of chills from art and music" has been formally accepted for publication in PLOS Genetics! Your manuscript is now with our production department and you will be notified of the publication date in due course.

With kind regards,

Judit Kozma

PLOS Genetics

On behalf of:
